# Phase Behavior of Gradient Copolymer Melts with Different Gradient Strengths Revealed by Mesoscale Simulations

**DOI:** 10.3390/polym12112462

**Published:** 2020-10-23

**Authors:** Pavel Beránek, Paola Posocco, Zbyšek Posel

**Affiliations:** 1Department of Informatics, Faculty of Science, Jan Evangelista Purkyně University in Ústí nad Labem, 40096 Ústí nad Labem, Czech Republic; pavelberanek91@gmail.com; 2Department of Engineering and Architecture, University of Trieste, 34127 Trieste, Italy; paola.posocco@dia.units.it

**Keywords:** nanomaterials, block copolymers, microphase separation, gradient copolymers, gradient strength, self-assembly, dissipative particle dynamics

## Abstract

Design and preparation of functional nanomaterials with specific properties requires precise control over their microscopic structure. A prototypical example is the self-assembly of diblock copolymers, which generate highly ordered structures controlled by three parameters: the chemical incompatibility between blocks, block size ratio and chain length. Recent advances in polymer synthesis have allowed for the preparation of gradient copolymers with controlled sequence chemistry, thus providing additional parameters to tailor their assembly. These are polydisperse monomer sequence, block size distribution and gradient strength. Here, we employ dissipative particle dynamics to describe the self-assembly of gradient copolymer melts with strong, intermediate, and weak gradient strength and compare their phase behavior to that of corresponding diblock copolymers. Gradient melts behave similarly when copolymers with a strong gradient are considered. Decreasing the gradient strength leads to the widening of the gyroid phase window, at the expense of cylindrical domains, and a remarkable extension of the lamellar phase. Finally, we show that weak gradient strength enhances chain packing in gyroid structures much more than in lamellar and cylindrical morphologies. Importantly, this work also provides a link between gradient copolymers morphology and parameters such as chemical incompatibility, chain length and monomer sequence as support for the rational design of these nanomaterials.

## 1. Introduction

Block copolymer self-assembly has been intensively studied and applied in material science for its ability to form highly ordered nanostructures [1]. Linear copolymer chains with various molecular weights and composition profiles (including diblock, triblock or multiblock copolymers; and random, taper or gradient copolymers) are now seen in many industrial applications [2]. Moreover, they are found in environment-sensitive applications, e.g., as bioactive molecular carriers, when used in solution [3,4,5] or self-healing materials [6].

Many of the above-mentioned applications rely on AB diblock copolymers, which consist of A and B blocks covalently bonded together with an abrupt change in the monomer density profile between the two blocks. Diblock copolymers self-assembly is controlled by three main parameters: the total polymerization of the chain N, the chemical incompatibility between the blocks described by the Flory-Huggins parameter $\chi_{AB}$, and the A/B block size ratio $f_{A}$. Increasing the chemical incompatibility above a critical value, ${\left(\chi_{AB}N\right)}_{c}$, triggers their assembly into various nanostructures with different dimensionality, representing their characteristic length. Lamellar (LAM) nanostructure has only one characteristic length: the distance between lamellae. Hexagonally packed cylinders (CYL) and spheres (SPH) ordered in a *bcc* or *fcc* lattice possess two characteristic lengths. These are the diameter of cylinders and the distance between them, and the diameter of spheres and lattice spacing, respectively. Finally, the gyroid (GYR) nanostructure, which belongs to the class of bicontinuous morphologies, exhibits two interpenetrating networks in all three dimensions.

Advances in polymer synthesis [6,7,8,9,10] have now allowed for the preparation of copolymers with controlled sequence chemistry [11]. For example, a new class of copolymer called taper has been prepared by inserting an additional block with a gradual change of A between the pure A and B blocks. [12,13] The taper block offers extra parameters to tailor the copolymer assembly, such as the length of the taper, orientation of the taper (normal or inverse) and type of taper (linear, random, etc.). Theoretical and experimental works showed that including the taper partly modifies the order-disorder transition, broadens the glass transition or widens the gyroid region. Nevertheless, in taper copolymers, only part of the chain is modified. Such chains represent a subclass of copolymers with controlled monomer composition along the chain. Copolymers with 100 % taper are called gradient copolymers. [14] Due to their monomer profile along the chain, gradient copolymers have found application in both melt and solution as filler dispersants [15], stabilizers and compatibilizers in immiscible blends [15,16] and as vibration and acoustic damping materials [17]. Examples of copolymers with different monomer sequences are shown in Figure 1.

Within the field of gradient copolymers, Aksimentiev and Hołyst [18] used the Ginzburg-Landau model to study gradient copolymers with various composition profiles, including linear, tan-h and exponential. They observed that melts with a linear composition profile exhibit only a transition from the disordered to the lamellar phase. Other phases known for diblock copolymers were found in melts with monotonic but non-linear profiles. Lefebvre et al. [19] applied Random Phase Approximation (RPA) and Self-Consistent Mean-Field (SCMF) theory to gradient copolymer melts with linear and tan-h composition profiles. They showed that melts with a linear profile phase separate at ${\left(\chi_{AB}N\right)}_{c}=29.25$, which is much higher than the common critical value for diblock copolymers ${\left(\chi_{AB}N\right)}_{c}=10.495$. Moreover, they demonstrated that in the strong-segregation limit (SSL) with $\chi_{AB}N=140$, the lamellar density profile of gradient copolymer melts remains sinusoidal, contrary to the abrupt change in the monomer density profile of diblock copolymers. Jiang et al. [20] used the SCMF framework and multiblock chain model with tan-h profile to obtain the phase diagrams (in $\chi_{AB}N-f_{A}\;$ plane) of melts with weak and strong gradient strength (i.e., the largest difference in monomer composition along the copolymer). They observed all nanostructures known for diblock copolymers and showed that the phase behavior of gradient copolymers is sensitive to the strength of the gradient profile. Melts with a strong gradient resemble the phase behavior of diblock copolymers, while gyroid and spherical nanostructures vanish in melts with weak gradients. Nevertheless, the authors argued that these phases, not present in the weak- and intermediate-segregation limit, might appear in SSL. Finally, for gradient copolymer melts with a linear profile only, the lamellar phase was predicted to be stable. Tito et al. [21] studied the lamellar nanostructures predicted in linear gradient copolymer melts by a combination of Self-Consistent Field (SCF) theory, scaling theory and predictions for SSL. They reported that the scaling of equilibrium lamellar spacing predicted for symmetric diblock copolymers $L_{eq}/R_{g}\;\sim {\left(\chi_{AB}N\right)}^{1/6}$ also holds for linear gradient copolymers. Mok et al. [22,23] prepared styrene/acrylic acid gradient copolymers and showed that these copolymers were much more efficient in reducing the interfacial tension than the corresponding diblock copolymers. Moreover, they showed that a monomer composition profile could be used to tune the glass transition. Ganesan et al. [24] studied the influence of monomer sequence polydispersity and blockiness on spinodal, phase behavior and the interfacial properties of gradient copolymer melts with linear and tan-h profiles. They used RPA for estimating the spinodal lines and Self-Consistent Brownian Dynamics (SCBD) to estimate the phase behavior. SCBD calculations were restricted to morphologies with two-dimensional symmetry, e.g., lamellae and hexagonally packed cylinders. They assessed that both the compositional polydispersity and blockiness of the sequence play a significant role in phase behavior. Larger influence was observed for systems with weak gradient strength. Jiang et al. [25] included polydispersity in their multiblock model, used previously for mono-sequence melts. Furthermore, they used RPA and SCFT to study the influence of monomer sequence polydispersity on phase behavior. An increase in polydispersity was shown to shift the order-disorder transition up and enlarge structures’ domain spacing.

Most theoretical and computational studies describing gradient copolymers rely on mean-field approximations, while the number of particle-based simulation studies is very limited and restricted to the lamellar phase or two dimensions [22,23,24,26]. For example, Pakula and Matyjaszewski [27] used the Monte Carlo method with a cooperative motion algorithm to study the lamellar phase. They considered random, block and gradient copolymers with different composition gradients. Sun et al. [28,29] applied Monte Carlo to investigate the interfacial properties and structure of ternary symmetric blends with gradient copolymers. Particle-based approaches naturally provide desired structure-property relationships, since they work on the scale of individual copolymer chains. Moreover, mesoscale simulations are ideal to understand the evolution and mechanisms of hierarchical self-assembling processes spanning multiple time and length scales, such as those involving gradient copolymers in melt, solution or a near-solid surface.

Therefore, the present study aims to provide a molecular insight into the phase behavior of gradient copolymer melts by using a particle-based simulation method known as Dissipative Particle Dynamics to derive the structure–property relationships required to rationally design these nanomaterials. To this end, we investigated the self-assembly of gradient copolymer melts with strong, intermediate, and weak gradient strengths. The scheme proposed by Fredrickson et al. [30], and later modified by Ganesan et al. [24], was adopted to model melts with a defined monomer density profile and a polydisperse monomer sequence. Phase behavior, ordering of individual phases and packing of chains was compared to those of the corresponding diblock copolymers for reference. To the best of our knowledge, this is the first study that provides molecular insight into the self-assembly of gradient copolymer melts by employing particle-based mesoscale modeling.

In the rest of the paper, we first describe the algorithm used to obtain melts with a polydisperse monomer sequence, defined monomer density profile and gradient strength. Then, the mesoscale chain model, simulation method and details are presented. Afterwards, we describe and discuss the results in the form of phase diagrams and a comparison of individual phases with different gradient strengths. Finally, we summarize the evidence in the Conclusions section. Additional information can be found in the Appendix A.

## 2. Mesoscopic Modelling 

In this section, we describe the algorithm employed to prepare melts with a defined tan-h monomer profile and a polydisperse monomer sequence. Then, we briefly describe the copolymer chain model adopted, simulation details and observables used to quantify the self-assembly and to identify equilibrium structures. Detailed discussion about proper identification of an equilibrium nanostructure related to the commensurability of structure and simulation box [31] is reported in the Appendix A.

### 2.1. Gradient Copolymer Melts with Polydisperse Monomer Sequence

Gradient copolymer melts, with specified monomer density profile f and a polydisperse monomer sequence, are here prepared with the method recently used by Ganesan et al. [24]. The polydispersity of the monomer sequence is controlled by the probability $p_{AA}$($p_{BB}$) that the segment A(B) is generated after the segment A(B) and is given by
(1)$$p_{AA}\left(i\right)=f\left(i\right)\left(1-\lambda \right)+\lambda$$
(2)$$p_{BB}\left(i\right)=f\left(i\right)\left(\lambda -1\right)+1.$$

Then, $p_{AB}=1-p_{AA}$ and $p_{BA}=1-p_{BB}$, respectively. The size of the blocks within the chain is determined by parameter λ. When λ→0, the blocks are small, while for λ→1, larger and more compact blocks are formed. Since Ganesan noted that the algorithm is most effective for λ ≲0.8, here we adopt λ=0.7. The monomer density profile f(i) (where *i* runs over all segments in the chain) is given by
(3)$$f\left(i\right)=\frac{1}{2}\left\{1+\tanh\left[C\pi \left(\frac{i}{N}-f^{*}\right)\right]\right\}$$
for which all the nanostructures known for diblock copolymers are predicted [20]. Gradient strength C determines the sharpness of the transition between A and B rich domains in the chain and represents a central parameter of this study. Melts with C→∞ correspond to diblock copolymers with an abrupt change in the monomer density profile, while melts with C→0 lead to completely random copolymers. $f^{*}$ is a parameter related to overall segment composition $\bar{f}$, such that
(4)$$\bar{f}=\frac{1}{N}\int_{1}^{N}f\left(i\right)di.$$

To satisfy the overall composition given by $\bar{f},$ Ganesan et al. [24] iteratively adjusted $f^{*}$. Here, we adopt a similar approach, where we first generate the chain with the monomer sequence given by Equations (1)–(3). Then, we calculate the overall composition $\bar{f}$ and, if the difference between calculated and target composition is less than ε (i.e., $\varepsilon =1e^{-4}$), we add the chain to our ensemble, otherwise we reject it. The parameter $f^{*}$ is modified only if a new chain is not accepted within 100 attempts. Nevertheless, 30 attempts are usually enough to generate an appropriate polydisperse monomer sequence. This approach enabled us to obtain melts with proper target monomer density profiles and polydisperse monomer sequences.

We consider melts with strong, C=5, intermediate, C={3,2}, and weak, C=1, gradient strength. The corresponding composition profiles considered here are shown for symmetric copolymers in Figure 1b. Appendix A shows the overall statistics for strong and weak gradient strengths and justify our choice of chain length N used in this study. In these figures, γ(i) represents the generated composition profile, $\phi_{b}\left(i\right)$ the block size distribution, and σ(i) the compositional polydispersity calculated as σ(i)=〈γ(i)−f(i)〉^2, all as a function of the scaled monomer position i in the chain. A comparison of the target f(i) and generated γ(i) profiles for different chain lengths is highlighted in Appendix A. Larger deviations are observed for weak gradient profiles, with N=60 being the shortest chain length that follows f(i) reasonably well. Despite chain length N=100 performing better, N=60 also meets reasonable computational criteria. Therefore, we adopted this chain length in our simulation study. A note must be made about the block size distribution $\phi_{b}\left(i\right)$ of copolymer with a weak gradient (C=1) in Appendix A. The shortest chain length (N=20) has a significantly different shape to the other distributions. This stems from the fact that we have fixed the first segment to be always A. As a result, due to short chain length and weak gradient strength, appropriately sized blocks could not be generated.

### 2.2. Dissipative Particle Dynamics and Gradient Copolymer Chain Model

Dissipative particle dynamics (DPD) is a well-established mesoscale simulation method that has been used several times for modelling polymers in melts [32,33], solutions or near surfaces [34,35], as well as the self-assembly of copolymers [36,37,38], nanocomposites [39,40] and out of equilibrium nanosystems [41]. Therefore, we refer the reader to reference [42] for full details on DPD and provide here only details relevant to our study. Additional information about DPD can be also found in ESI.

For our DPD simulations, we adopt standard reduced units. $k_{B}T$ is the unit of energy, where $k_{B}$ is the Boltzmann constant and T the thermodynamic temperature. Cutoff distance $r_{c}$ and mass m of the bead are the unit of length and mass, respectively. All beads in our model have same mass and volume. The total reduced bead density is set to $\rho r_{c}^{3}=3$. All beads interact with standard DPD potential, where $\left(a_{AB}r_{c}\right)/\left(k_{B}T\right)$ is the maximum repulsion between unlike beads and is related to the standard Flory-Huggins interaction parameter $\chi_{AB}$ [43]. The Mesoscale model of gradient copolymer chain consists of $N_{AB}=N_{A}+N_{B}=60$ beads, with the ratio of A segments in the chain given by $f_{A}=N_{A}/N_{AB}$. Adjacent beads in the chain are bonded with a spring described by the force
(5)$$f_{i,i+1}^{bond}=K_{s}\left(r_{i,i+1}-r_{0}\right)$$
where $K_{s}=4k_{B}T$ is the stiffness of the spring, and $r_{\left(i,i+1\right)}=\left|r_{i+1}-r_{i}\right|$ and $r_{0}=0r_{c}\;$ are the equilibrium distances of the spring.

### 2.3. Simulation Details

In addition to the visual inspection of obtained configurations, we also measure several structural characteristics. To distinguish equilibrium structures and set the proper simulation box length [44], we calculate the structure factor S(q) as
(6)$$S\left(q\right)=\;\frac{1}{N}\left[{\left(\overset{N}{\underset{i=1}{\sum }}\cos\left(q\cdot r_{i}\right)\right)}^{2}+{\left(\overset{N}{\underset{i=1}{\sum }}\sin\left(q\cdot r_{i}\right)\right)}^{2}\right]$$
where q is the wave vector, $r_{i}$ is the position of i-th segment in the simulation box and N runs over all segments that form the structure. The knowledge of S(q) allows us to identify the unit cell of equilibrium structure $L_{unit}$ and set the proper box dimensions as
(7)$$L_{unit}=\frac{2\pi }{q^{*}}m\;\to L_{box}=nL_{unit}$$
where $q^{*}$ is the position of the first maxima in S(q), $m=q_{2}/q^{*}$ is a number specific to each type of structure and n is a multiple of the unit cell. For example, a lamellar nanostructure has m=1, gyroid has $m=\sqrt{4}/\sqrt{3}$, hexagonally packed cylinders have $m=\sqrt{3}$ and a spherical bcc nanostructure has $m=\sqrt{2}$ [1].

To describe the structure’s degree of ordering, we further calculate the order parameter $P_{OP}$ as
(8)$$P_{OP}=\frac{1}{V}\int_{V}^{}\left[\rho_{i}^{2}\left(r\right)-\rho_{i}^{2}\right]dV$$
where $\rho_{i}^{2}\left(r\right)$ is the squared local density of type i={A,B} at position r and $\rho_{i}^{2}$ is the squared overall density of type i in the system. The $P_{OP}$ approaches zero for completely disordered systems and approaches the target number (different for each type of structure) for ordered configurations. Finally, to describe the packing of individual chains, we calculate chains’ mean-squared radius-of-gyration $R_{g}^{2}$.

All initial configurations and post-processing tools are prepared by in-house codes developed in Python and FORTRAN language. Simulations are performed in LAMMPS [45] with a GPU package [46]. All calculations start with an initial cubic box size equal to $L=40r_{c}$, containing a total number of beads equal to $N=L^{3}\rho r_{c}^{3}=192000$ and a total number of chains equal to $n_{c}=3200$. The simulation step is set to ∆t=0.05, the friction coefficient to γ=4.5 and the cutoff distance to $r_{c}=1.0$.

Simulations start from a random initial configuration equilibrated with $a_{AA}=a_{AA}=a_{AB}=25\left(k_{B}T\right)/r_{c}$. Then, the interaction $a_{AB}$ is gradually increased with $∆a_{AB}=2\left(k_{B}T\right)/r_{c}$ up to the value where we observe complete phase separation. Each increase in $a_{AB}$ is followed by additional $1\;\times \;10^{6}$ simulation steps, and the order parameter $P_{OP}$ is measured. Plateau in $P_{OP}$ indicates that complete separation is reached and ordered structure is formed. If the structure is labeled as equilibrium one, additional $1\;\times \;10^{3}$ configurations are collected during the subsequent $1\;\times \;10^{6}$ simulation steps to calculate ensemble averages. Similar approach was used before by one of us to model the phase behavior of semiflexible-flexible diblock copolymer melt [47]. Graphical workflow of the above described approach, together with the description of the two-step method for identification of equilibrium nanostructures, are presented in detail in the Appendix A (see, for example, Appendix A). Within this framework, obtaining each column in the phase diagram required approximately 2 days of calculation using single-GPU, together with subsequent CPU analysis of the results.

## 3. Results and Discussion

To validate our two-step method for finding equilibrium structures, we first derived the phase diagram of diblock copolymers. Then, we presented complete phase diagrams for gradient copolymer melts with strong, intermediate and weak gradient strengths. The influence of polydisperse monomer sequences on the formation and stability of individual phases is discussed by plotting the distribution of A/B block size ratio $f_{A}$, order parameter $P_{OP}$, and mean-squared radius-of-gyration $R_{g}^{2}$.

The complete phase diagram of diblock copolymers is shown in Appendix A in ESI. Only half of this symmetric diagram is displayed in the $\chi_{AB}N-\bar{f}$ plane. The relation between bead–bead interaction in parameter $a_{AB}$ and Flory-Huggins parameter $\chi_{AB}$ is shown in the Equation (S2) in ESI. Filled circles represent simulation points where we observe complete phase separation and equilibrium structure. Open circles then represent points where the system remains disordered. Different colors display different structures, where red circles denote lamellar, green circles gyroid, blue circles hexagonally packed cylinders and violet circles a spherical structure, respectively. Black dashed lines highlight approximate phase boundaries between individual phases and the order/disorder region. Fittingly, the expected structures and their stability regions are well detected and in perfect agreement with those published before [48]. 

Figure 2 reports the phase diagrams of gradient copolymer melts with strong, intermediate and weak gradient strength in the $\chi_{AB}N-\bar{f}$ plane. For direct comparison, the dashed line in Figure 2 represents the approximate phase boundaries of the diblock copolymers. Comparison between diblock and gradient melts with strong gradients (C=5) in Figure 2a shows that phase boundaries are only marginally influenced. This is consistent with previous theoretical phase diagrams presented, for example, by Jiang et al. [20]. The influence of the gradient part is more pronounced for melts with higher $\bar{f},$ while almost no effect is observed for melts closer to symmetric copolymers, $\bar{f}=0.5$.

In addition, Figure 3 displays the distribution $N\left(f_{A}\right)$ of the A/B block size ratio around the overall distribution $\bar{f}$ determined by diblock copolymers. Distributions for diblock, strong and weak gradient melts with $\bar{f}=0.5$ (Figure 3a), $\bar{f}=0.7$ (Figure 3b), and $\bar{f}=0.9$ (Figure 3c) are shown. We see that the majority of chains in strong gradient melts (Figure 2a) are close to the overall ratio $\bar{f}$ (red bars), where a lamellar structure is formed. Increasing $\bar{f}$ to 0.7 (Figure 3b) leads to the presence of copolymer chains with compositions closer to homopolymer, and higher A/B incompatibility is required to assemble such chains into ordered structures. Consequently, formation of the gyroid phase and hexagonally packed cylinders shown in Figure 2a is observed at higher $\chi_{AB}N$ values. For $\bar{f}$ equal to 0.9, where we expect spherical domains, we see that gradient melts contain high amounts of homopolymers (Figure 3c). For example, a gradient melt with strong gradient strength contains almost 40% homopolymers. Therefore, much higher incompatibility is required to drive such chains into ordered spherical domains or these domains are not able to form.

Decreasing gradient strength to intermediate values C={2,3} leads to the widening of the gyroid phase window at the expense of the hexagonally packed cylinders in the phase diagrams of Figure 2b,c. Brown et al. [49,50,51] reported the same effect for taper block copolymers. Within the framework of SCFT and RPA (and later confirmed by molecular dynamics calculations), they assessed that the size and length of tapers have significant influence on the formation/position of ordered phases. Increasing the length of the taper significantly increases the ${\left(\chi_{AB}N\right)}_{c}$ and leads to a wider gyroid phase window with respect to diblock copolymers, as well as to a slight shift of curved phases to smaller $f_{A}$ values. Here, due to the high content of homopolymer chains, spherical domains are shifted to high $\chi_{AB}N$ values. Only minor differences, including formation of the lamellar phase at the order-disorder transition for C=2, are discerned between C=3 and C=2  phase diagrams.

For weak gradient strength (C=1), the monomer density profile is close to a linear shape (Figure 1b) and only the lamellar structure is expected, which indeed is the only stable phase detected for C=1 in Figure 2d. The lamellar phase extends here up to $\bar{f}=0.75$. Such extension can be explained by considering that the A/B block size distribution in Figure 3 (black bars) is much broader when compared with C=5. Therefore, at $\bar{f}=0.7,$ the majority of chains in gradient melts with weak gradients are still able to form lamellar phases. Further increase of $\bar{f}$ leads to the disorder phase, although we may speculate that other phases might appear at higher $\chi_{AB}N$ values. Nevertheless, we did not see them when we increased incompatibility above $\chi_{AB}N=300$, well beyond the boundary of our phase diagrams.

The influence of gradient strength on individual phases is described by the order parameter $P_{OP}$ (Figure 4a) and the chains’ mean-squared radius-of-gyration $R_{g}^{2}$ (Figure 4b). The order parameter calculated by Equation (8) is shown for the lamellar, gyroid and cylindrical phases. We see that decrease in gradient strength is followed by a monotonic decrease in the ordering of all phases. Non-monotonic change in $P_{OP}$ is observed for the lamellar phase at a weak gradient strength, where the monomer density profile approaches a linear shape. Corresponding snapshots in Figure 4c and Appendix A evidence that lowering gradient strength decreases A/B interface tension and promotes the mixing of A and B phases. At the lowest gradient strength considered here (C=1), the A/B interface is hardly distinguishable (see Appendix A). The conformational behavior of individual chains at different gradient strengths is reported in Figure 4b. We see that while chains’ stretching in the lamellar phase is similar for strong and intermediate gradient strength, the smeared A/B interface allows chains to adopt more packed conformations in lamellar phases with weak gradient strength, where $R_{g}^{2}$ decreases. Similar behavior also occurs for hexagonally packed cylinders. On the other hand, gyroid morphologies appear more sensitive to variations in gradient strength. We think this is strictly related to the peculiar structure of the gyroid phase, where the interconnected network forces the chains to adopt stretched conformations, especially at high C values where the interface tension among the A/B phase is at a maximum. Decreasing the gradient strength relieves this confinement, allowing for chain relaxation and an increase in packed conformations.

## 4. Conclusions

In this work, we apply dissipative particle dynamics for predicting the phase behavior of gradient copolymer melts with different gradient strengths. We consider melts with strong, intermediate, and weak gradient strengths and compare their phase behavior to that of corresponding diblock copolymers. Gradient melts with polydisperse monomer sequences are modelled, and a two-step method for finding equilibrium nanostructures is applied, including the scaling of box dimensions and the application of temporal shear flow.

The results show that gradient melts with a strong gradient resemble diblock copolymers, with only minor changes related to order-disorder phase transitions that are far from symmetric compositions. Moreover, we assess that decreasing the gradient strength to intermediate values leads to a wider gyroid window at the expense of the cylindrical phase. This was also reported before for tapered copolymers with tapers increasing from 30% to 50%. Due to polydisperse monomer sequences, melts can contain larger amounts of homopolymers that shift the order-disorder transition to higher values (compared with diblock copolymers), especially in the spherical phase, where high content of homopolymers may even prevent its formation. Furthermore, the phase behavior of gradient melts with weak gradients exhibits only the lamellar phase, suppressing the formation of other morphologies. Monomer density profiles close to linear ones either shift the order-disorder transitions of the other phases to very high A/B incompatibility values or even suppresses their formation due to the broad distribution of the A/B  block size ratio and high content of homopolymer chains. Finally, we demonstrate that decreasing the gradient strength relieves  A/B interface tension, promotes the mixing of A and B phases, and allows chains to adopt more packed conformations.

Overall, we establish here that dissipative particle dynamics, coupled with mesoscale description of chains with polydisperse monomer sequence, are able to capture the phase behavior of gradient copolymer melts successfully. Significantly, they allow for the retrieval of fundamental relationships connecting key parameters controlling self-assembly with overall structural properties as well as individual chain characteristics. We believe that the complete phase space discussed in this work will constitute an essential tool for the rational design of these nanomaterials. Results and methods illustrated here also open the way for the exploration of more complex systems such as gradient copolymers in solutions or films.

## Figures and Tables

**Figure 1 polymers-12-02462-f001:**
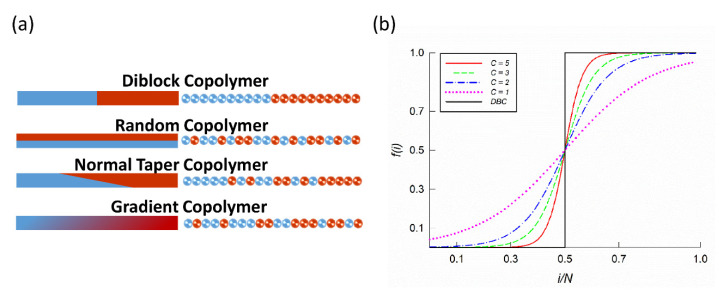
(**a**) Examples of copolymers with different monomer sequences along the chain. The left side shows bars with monomer density profiles and the right side displays possible monomer sequences in the chain. (**b**) Monomer composition profile f(i) for the A segment in a copolymer chain with total length N. Figure shows all tan-h composition profiles considered here, with different gradient strengths (C={1,2,3,5}) and composition profiles of corresponding diblock copolymers (DBC).

**Figure 2 polymers-12-02462-f002:**
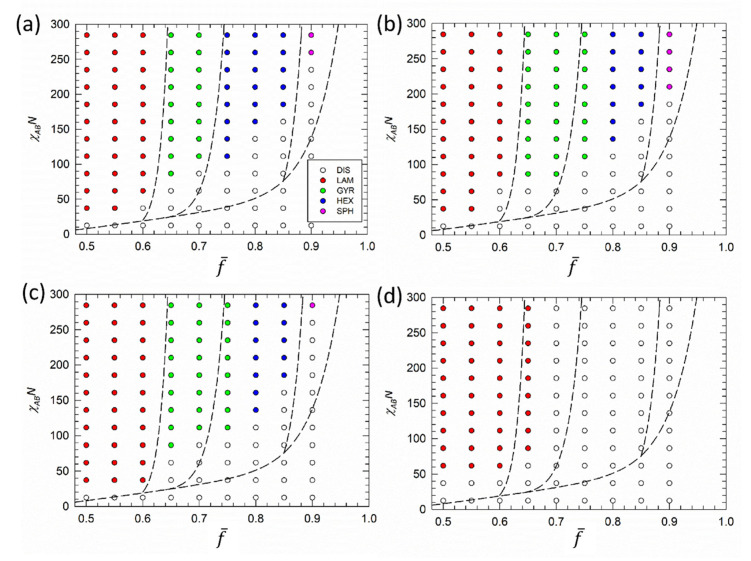
Gradient copolymer phase diagrams. Melts with C = {(**a**) 5, (**b**) 3, (**c**) 2, (**d**) 1)} are shown in the $\chi_{AB}N-\bar{f}$ plane, where $\chi_{AB}$ is the Flory-Huggins interaction parameter between unlike beads and $\bar{f}$ is the fraction of A segments in the copolymer chain. Symbols represent simulation points, where red circle stands for lamellae, green circle for gyroid, blue circles for hexagonally packed cylinders, and pink circles for spherical nanostructures, respectively. Open circles represent the disordered phase. Black dashed lines denote approximate phase boundaries of corresponding diblock copolymers.

**Figure 3 polymers-12-02462-f003:**
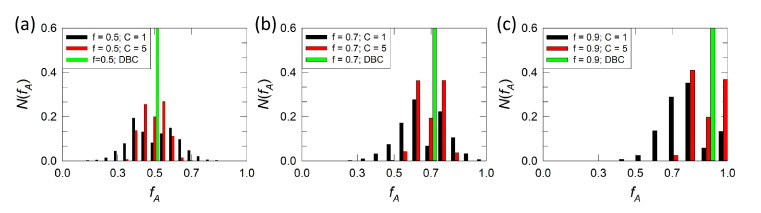
Distribution of chains $N\left(f_{A}\right)$ as a function of the fraction of A segments that become $f_{A}$ in copolymer chains for (**a**) $\bar{f}=0.5$, (**b**) $\bar{f}=0.7$ and (**c**) $\bar{f}=0.9$. Only gradient copolymer melts with C={5,1} are shown by black and red bars, respectively. Green bar denotes corresponding diblock copolymers with $N\left(f_{A}\right)=1$.

**Figure 4 polymers-12-02462-f004:**
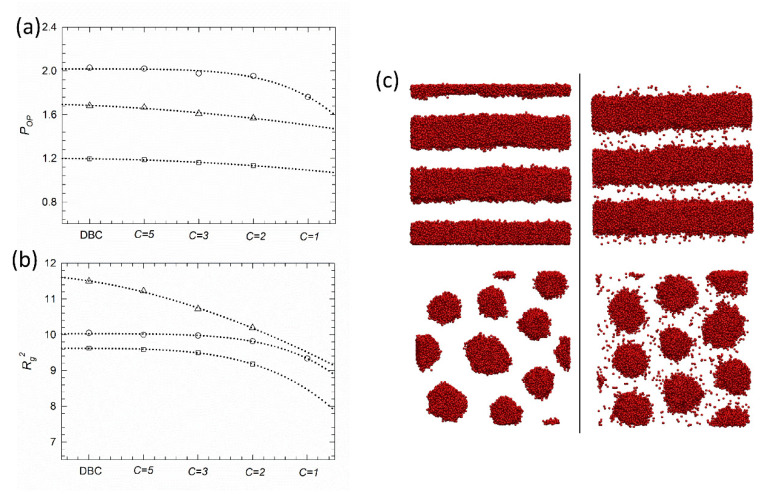
(**a**) Order parameter $P_{OP}$ and (**b**) chain mean-squared radius-of-gyration $R_{g}^{2}$ for the lamellar phase (LAM, $\bar{f}=0.5$, open circle), hexagonally packed cylinders (CYL, $\bar{f}=0.8$, open square), and the gyroid phase (GYR, $\bar{f}=0.5$, open triangle) for all the different melts considered here. Error bars within size of the symbol are not shown. (**c**) Configurational snapshots of lamellar (top) and front view of hexagonally packed cylinder (bottom) assemblies with $\left(a_{AB}r_{c}\right)/\left(k_{B}T\right)=40$. Snapshots of diblock copolymers and gradient copolymers with C=2 are shown in left and right column, respectively. For clarity, only A segments are displayed. Additional snapshots are reported in the Appendix A.

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
