# Peer review of "Phase Behavior of Gradient Copolymer Melts with Different Gradient Strengths Revealed by Mesoscale Simulations"

_polymers, 2020, doi:10.3390/polym12112462_

Round 1

Reviewer 1 Report

Dissipative particle dynamics calculations have been done to predict the phase behavior of different copolymer melts. It is considered melts with different gradient strength. This study is useful to understand and to design nanomaterials with specific microscopic structure. The system and simulations are well explained. The authors use a two-step method for finding equilibrium structures. However, this seems to be a limitation of the selected size of the simulations. Bigger sizes should be enough to arrive directly to the equilibrium properties. It should be interesting to determine the magnitude of such dimensions.

-It should be interested to indicate in the manuscript the time required to perform a typical run of some of the system.

-Line 191, r0=0rc should be corrected.

-In the final paragraph of part 2 and in the ESI, the scientific notation is not well expressed, as in 1e6, 1e3…

- The size of the labelling and numbers in Figure 3 should be increased to be more readable.

-The electronic supplementary information should be given in pdf format to be compatible with any system or version.

Author Response

We thank the reviewer for her/his time and expertise to perform this review and for valuable comments and corrections. All changes done in the manuscript are marked with red color.

Dissipative particle dynamics calculations have been done to predict the phase behavior of different copolymer melts. It is considered melts with different gradient strength. This study is useful to understand and to design nanomaterials with specific microscopic structure. The system and simulations are well explained. The authors use a two-step method for finding equilibrium structures. However, this seems to be a limitation of the selected size of the simulations. Bigger sizes should be enough to arrive directly to the equilibrium properties. It should be interesting to determine the magnitude of such dimensions.

Unfortunately, reaching the equilibrium nanostructure is not a problem of system size but of a periodic nanostructure that assembles in simulation box with periodic boundary conditions. Then, the dimensions of the structure must commensurate well with dimensions of the simulation box otherwise long-lived metastable states are observed. Having that in mind and from our previous experience (see for example paper Skvor, J.; Posel, Z. Simulation Aspects of Lamellar Morphology: Incommensurability Effect. Macromol Theor Simul 2015, 24, 141-151.), we have designed this two-step method to handle this issue effectively.

- It should be interested to indicate in the manuscript the time required to perform a typical run of some of the system.

Calculation time depends on two major things. First, the simulation scheme used to obtain the equilibrium configuration (structure factor or shear flow), second the hardware. Here we used the GPU GTX 2080Ti which was a state-of-the-art GPU machine last year. We have added a rough estimation of time to obtain one column in phase diagram at the end of the Simulation Details section.

-Line 191, r0=0rc should be corrected.

In the original DPD method the equilibrium distance of the spring bond is indeed set to zero. This number changes for targeting specific systems. Since we used a generic DPD model, we left also the original setting of r0.

-In the final paragraph of part 2 and in the ESI, the scientific notation is not well expressed, as in 1e6, 1e3…

Corrected.

- The size of the labelling and numbers in Figure 3 should be increased to be more readable.

Increased.

-The electronic supplementary information should be given in pdf format to be compatible with any system or version.

The SI file is now provided as a pdf.

Reviewer 2 Report

In this work dissipative particle dynamics is used to predict the phase behavior of gradient copolymer melts with different gradient strengths. While the work is theoretical in natural, it does an very good job providing insight into the phase behavior of copolymer melts. Such insight is invaluable to design processes and future studies. The work was well planned and executed, and the paper is very well written to clear articular their work and findings.

I would additionally like to commend the authors for their use of supporting information. The material provided is thorough and may be of interest to other similar researchers. However, by moving this paper to the supporting information, the manuscript reads beautifully. I offer just three very minor comments.

  1. On line 217 you use "1e6", where 6 is superscript, to indicate the number of MC steps. I would recommend using standard scientific notation to avoid confusion. The same is true on line 219.
  2.  The simulations were provided with LAMMPS, which is open-source. In the supporting information, please provide example LSAMMPS input files that could be used to reproduce your results. This could eliminate potential questions, and ensure your work to be reproducible by others.
  3. While I trust this will be taken care of by the editors, please make the supporting information document a PDF. I do not use Microsoft Word, and on my computer some of the formatting was off.

Author Response

In this work dissipative particle dynamics is used to predict the phase behavior of gradient copolymer melts with different gradient strengths. While the work is theoretical in natural, it does an very good job providing insight into the phase behavior of copolymer melts. Such insight is invaluable to design processes and future studies. The work was well planned and executed, and the paper is very well written to clear articular their work and findings.

I would additionally like to commend the authors for their use of supporting information. The material provided is thorough and may be of interest to other similar researchers. However, by moving this paper to the supporting information, the manuscript reads beautifully. I offer just three very minor comments.

We thank the reviewer for her/his appreciation and for valuable comments and corrections. All changes done in the manuscript are marked with red color.

  1. On line 217 you use "1e6", where 6 is superscript, to indicate the number of MC steps. I would recommend using standard scientific notation to avoid confusion. The same is true on line 219.

Corrected.

  1.  The simulations were provided with LAMMPS, which is open-source. In the supporting information, please provide example LSAMMPS input files that could be used to reproduce your results. This could eliminate potential questions, and ensure your work to be reproducible by others.

Two simulation schemes were added to the ESI and the reference was added to the appropriate figures.

  1. While I trust this will be taken care of by the editors, please make the supporting information document a PDF. I do not use Microsoft Word, and on my computer some of the formatting was off.

The SI file is now provided as a pdf.